# Clinical and Diagnostic Utility of Genomic Profiling for Digestive Cancers: Real-World Evidence from Japan

**DOI:** 10.3390/cancers16081504

**Published:** 2024-04-15

**Authors:** Marin Ishikawa, Kohei Nakamura, Ryutaro Kawano, Hideyuki Hayashi, Tatsuru Ikeda, Makoto Saito, Yo Niida, Jiichiro Sasaki, Hiroyuki Okuda, Satoshi Ishihara, Masatoshi Yamaguchi, Hideaki Shimada, Takeshi Isobe, Yuki Yuza, Akinobu Yoshimura, Hajime Kuroda, Seigo Yukisawa, Takuya Aoki, Kei Takeshita, Shinichi Ueno, Junichi Nakazawa, Yu Sunakawa, Sachio Nohara, Chihiro Okada, Ko Nishimiya, Shigeki Tanishima, Hiroshi Nishihara

**Affiliations:** 1Genomics Unit, Keio Cancer Center, Keio University School of Medicine, Integrated Medical Research Building 3-S5, 35, Shinanomachi, Shinjuku-ku, Tokyo 160-8582, Japan; knakamura320@keio.jp (K.N.); ryu.kawano@keio.jp (R.K.); rock-hayashi-pop@keio.jp (H.H.); tanishima.shigeki@gmail.com (S.T.); hnishihara1971@keio.jp (H.N.); 2Department of Cancer Genome Medical Center, Hakodate Goryoukaku Hospital, 38-3, Goryoukakucho, Hakodate-shi 040-8611, Hokkaido, Japan; tatsuruikeda@gmail.com; 3Department of Genetic Medicine, Ibaraki Prefectural Center Hospital, 6528, Koibuchi, Kasama-shi 309-1793, Ibaraki, Japan; ma-saito@chubyoin.pref.ibaraki.jp; 4Center for Clinical Genomics, Kanazawa Medical University Hospital, 1-1, Daigaku, Uchinada 920-0293, Ishikawa, Japan; niida@kanazawa-med.ac.jp; 5Research and Development Center for New Medical Frontiers, Kitasato University School of Medicine, 1-15-1 Kitasato, Minami-ku, Sagamihara-shi 252-0329, Kanagawa, Japan; saji@med.kitasato-u.ac.jp; 6Department of Medical Oncology, Keiyukai Sapporo Hospital, 1-1 Minami, Hondori 9, Chome, Shiroishi-ku, Sapporo 003-0026, Hokkaido, Japan; hiokuda@keiyukaisapporo.or.jp; 7Cancer Genome Diagnosis and Treatment Center, Central Japan International Medical Center, 1-1 Kenkonomachi, Minokamo-shi 505-0010, Gifu, Japan; s-ishi@cjimc-hp.jp; 8Division of Clinical Genetics, Faculty of Medicine, University of Miyazaki Hospital, 5200 Kihara, Kiyotake-cho, Miyazaki-shi 889-1692, Miyazaki, Japan; myama@med.miyazaki-u.ac.jp; 9Department of Surgery and Clinical Oncology, Toho University Graduate School of Medicine, 6-11-1 Omori-nishi, Ota-ku, Tokyo 143-8541, Japan; hideaki.shimada@med.toho-u.ac.jp; 10Cancer Genome Medical Center, Shimane University Hospital, 89-1, Enya-cho, Izumo-shi 693-8501, Shimane, Japan; isobeti@med.shimane-u.ac.jp; 11Department of Hematology and Oncology, Tokyo Metropolitan Children’s Medical Center, 2-8-29 Musashidai, Fuchu-shi 183-8561, Tokyo, Japan; yuki_yuza@tmhp.jp; 12Department of Clinical Oncology Director, Outpatient Chemotherapy Center, Tokyo Medical University Hospital, 6-7-1 Nishishinjuku, Shinjuku-ku, Tokyo 160-0023, Japan; gangeno@tokyo-med.ac.jp; 13Department of Pathology, Tokyo Women’s Medical University, Adachi Medical Center, 4-33-1 Kohta, Adachi-ku, Tokyo 123-8558, Japan; kuroda.hajime@twmu.ac.jp; 14Department of Medical Oncology, Saiseikai Utsunomiya Hospital, 911-1, Takebayashi, Utsunomiya-shi 321-0974, Tochigi, Japan; seigo_yukisawa@saimiya.com; 15Department of Clinical Oncology, Tokyo Medical University Hachioji Medical Center, 1163, Tatemachi, Hachioji-shi 193-0998, Tokyo, Japan; takuaoki@tokyo-med.ac.jp; 16Department of Clinical Genetics, Tokai University Hospital, 143, Shimokasuya, Isehara-shi 259-1193, Kanagawa, Japan; takeshita_kei@mac.com; 17Oncology Center, Kagoshima University Hospital, 8-35-1 Sakuragaoka, Kagoshima-shi 890-0075, Kagoshima, Japan; ueno1@m.kufm.kagoshima-u.ac.jp; 18Department of Medical Oncology, Kagoshima City Hospital, 37-1, Uearatacho, Kagoshima-shi 890-8760, Kagoshima, Japan; nakazawa-j87@kch.kagoshima.jp; 19Department of Clinical Oncology, St. Marianna University School of Medicine, 2-16-1 Sugao, Miyamae-ku, Kawasaki 216-8511, Kanagawa, Japan; y.sunakawa@marianna-u.ac.jp; 20Biomedical Informatics Department, Communication Engineering Center, Mitsubishi Electric Software Corporation, Fuji Techno-Square, 5-4-36 Tsukaguchi-Honmachi, Amagasaki-shi 661-0001, Hyogo, Japan; nohara.sachio.sl@mesw.co.jp (S.N.); okada.chihiro.kh@mesw.co.jp (C.O.); nishimiya.ko.dc@mesw.co.jp (K.N.)

**Keywords:** adenocarcinoma, cancer gene panel, comprehensive genomic profiling, pathological diagnosis

## Abstract

**Simple Summary:**

The clinical and diagnostic utility of comprehensive genomic profiling (CGP) in Japan has not been thoroughly investigated. To address this gap, this large-scale study aimed to determine the usefulness of CGP in diagnosing digestive cancer. A total of 547 cases of digestive cancers were analyzed using an original scoring system. Through this approach, the characteristic genomic profiles of each digestive cancer type were identified, with the presence or absence of *APC*, *KRAS*, and *CDKN2A* alterations being characteristic of each organ. Based on the patterns of genomic alterations characteristic of each digestive cancer type, we suggested a classification flowchart specifically designed for digestive adenocarcinomas. Our findings highlight not only the clinical utility of CGP but also its diagnostic utility for digestive cancers.

**Abstract:**

The usefulness of comprehensive genomic profiling (CGP) in the Japanese healthcare insurance system remains underexplored. Therefore, this large-scale study aimed to determine the usefulness of CGP in diagnosing digestive cancers. Patients with various cancer types recruited between March 2020 and October 2022 underwent the FoundationOne^®^ CDx assay at the Keio PleSSision Group (19 hospitals in Japan). A scoring system was developed to identify potentially actionable genomic alterations of biological significance and actionable genomic alterations. The detection rates for potentially actionable genomic alterations, actionable genomic alterations, and alterations equivalent to companion diagnosis (CDx), as well as the signaling pathways associated with these alterations in each digestive cancer, were analyzed. Among the 1587 patients, 547 had digestive cancer. The detection rates of potentially actionable genomic alterations, actionable genomic alterations, and alterations equivalent to CDx were 99.5%, 62.5%, and 11.5%, respectively. *APC*, *KRAS*, and *CDKN2A* alterations were frequently observed in colorectal, pancreatic, and biliary cancers, respectively. Most digestive cancers, except esophageal cancer, were adenocarcinomas. Thus, the classification flowchart for digestive adenocarcinomas proposed in this study may facilitate precise diagnosis. CGP has clinical and diagnostic utility in digestive cancers.

## 1. Introduction

The advent of next-generation sequencing (NGS) has enabled the comprehensive and rapid analysis of genomic information at low costs. Comprehensive cancer gene analysis, defined as the simultaneous analysis of a large number of cancer-related genes in a single test, has gained popularity in recent years. Among the estimated 20,000 genes present in the human genome, approximately 400 are cancer-related genes [1]. Cancer-related genes have been analyzed using NGS-based cancer gene panels to detect genomic alterations. The process of testing cancer-related genes using a cancer gene panel is known as comprehensive genomic profiling (CGP), as it yields a comprehensive genomic profile of the cancer. CGP has become an indispensable test in the domain of cancer genome medicine [2,3,4]. Tumors have been assessed using NGS-based CGP to detect genomic alterations, such as base substitutions, insertions and deletions (indels), and fusions/rearrangements. Copy number alterations (CNAs) [5], a part of precision cancer medicine, have been used to identify effective targeted therapies that consider individual genomic variability and susceptibility. CGP testing of tissue samples using the FoundationOne^®^ CDx assay (Foundation Medicine, Cambridge, MA, USA) was approved for insurance coverage in Japan in June 2019 [6]. 

FoundationOne^®^ CDx, a CGP platform that exclusively examines tumor tissue, focuses on a targeted sequence capable of analyzing the hotspots of 324 genes, including 36 fusion genes [7]. FoundationOne^®^ CDx also possesses companion diagnostic (CDx) functions in multiple molecular-targeted therapeutics for specific genes [8]. Its use has enabled the application of NGS to in vitro diagnostics via a hybrid capture-based target enrichment approach and the construction of a whole-genome shotgun library for the identification of substitutions, indels, CNAs, and select rearrangements, the four classes of somatic genomic alterations [9]. Formalin-fixed paraffin-embedded (FFPE) specimens that are 5 × 5 mm in size, have a tumor content of at least 20%, and have been collected within the last 3 years are recommended. FoundationOne CDx, the representative CGP in Japan, accounts for 74% of the CGP performed by our group up to October 2022 [based on unpublished data from the authors’ facility].

Previous studies have examined the clinical utility of CGP tests covered by the Japanese public health insurance system mainly from the clinical perspective, including genomic profiling in light of pharmacotherapy and other treatments [10,11,12,13]. However, the correlation of genomic profiles with pathological features and the signaling pathways involved in digestive cancers remain underexplored. The genomic profiles contain many kinds of diagnostically significant information that describes the nature of the tumor, and CGP is a necessary test for qualitative diagnosis of tumors. In particular, the WHO classification gives diagnoses to most tumors according to their genomic alterations for hematological [14], brain [15], and bone and soft tissue tumors [16]. Therefore, genomic analysis is essential when making a diagnosis.

This study aimed to characterize the pathological genomic features of different types of digestive cancers in clinical practice using a CGP with insurance coverage and establish an original scoring system. The information obtained using CGP will aid in the regularization of its use in routine clinical practice across Japan and facilitate the creation of a database for optimizing treatment strategies for cancer. 

## 2. Materials and Methods

This study was approved by the Ethics Committee of the Keio University School of Medicine (approval number: 2021-1159). Between March 2020 and October 2022, 1587 patients with cancer were selected for this study. These patients underwent the FoundationOne CDx [7] assay at one of the hospitals affiliated with the Keio PleSSison Group, the Keio University Hospital (a core hospital for cancer genome medicine recognized by the Japanese Ministry of Health, Labour and Welfare) and its 18 partner hospitals.

CGP was performed under the coverage of the Japanese insurance system. The requirements for insurance reimbursement are as follows: (1) patients, excluding those with hematologic tumors, who have completed or were expected to complete standard medical therapy, and (2) patients with an unknown primary cancer or rare cancer with no established treatment protocol. 

Appendix A presents the clinical workflow of the cancer genome testing in the Keio PleSSision Group. Consent was obtained via the opt-out method at the outpatient clinic of each hospital. The tumor specimens were sequenced at Foundation Medicine, Inc. (FMI) (Cambridge, MA, USA) and curated in bulk by the Molecular Tumor Board at the Genomics unit, Keio University, based on reports analyzed by the Center for Cancer Genomics [17] and Mitsubishi Electric Software Corporation (Amagasaki-shi, Hyogo, Japan). The Clinical Tumor Board of each hospital reviewed the treatment recommended based on the sequencing results. The results were explained to the patients in the outpatient clinic at each hospital after a web-based consultation [18].

### 2.1. Sequencing and Identification of Genomic Alterations

DNA extraction and data acquisition via sequencing were performed in accordance with predefined protocols followed at the facilities designated by the FMI. A total of 324 genes were sequenced via NGS using FoundationOne CDx. The presence of genomic alterations, such as base substitution, indels, fusions/rearrangements, and CNAs, were evaluated. CGP was performed as described in a previous study [8]. Although corrected to some extent by tumor content, a variant allele frequency of 10% was established as the cut-off value.

An original scoring system was developed based on the following factors to evaluate and score the genomic alterations: population of carcinoma clones, the function of gene alteration (using reference databases such as COSMIC, ClinVar, OncoKB, CIViC, and JAX CKB), and the effect of CNAs (Figure 1). The total score of these three categories was defined as the alteration score. Tumor mutation burden (TMB) and microsatellite instability (MSI) status were also evaluated [5,19]; high TMB (TMB-H) was defined as the presence of ≥10 single-nucleotide variants/Mbp. A scoring system for genomic alterations was developed based on these parameters.

“Potentially actionable genomic alterations” were defined as genomic alterations with a biological significance of ≥2 but no variants of unknown significance (VUSs). “Actionable genomic alterations” were defined as genomic alterations eligible for drug development with a potential usefulness of ≥2.5; an evidence level of D or higher according to the guidance provided by the Japanese Society of Medical Oncology, Japanese Society of Clinical Oncology, and Japanese Cancer Association; and without VUSs (Figure 1) [20].

“Genomic alterations equivalent to CDx” were defined as genomic alterations with which the physician could use the specific drugs under insurance coverage in Japan (Appendix A). The detection rate of potentially actionable genomic alterations, actionable genomic alterations, and genomic alterations equivalent to CDx, as well as the signaling pathways activated or inactivated by genomic alterations, were evaluated for each type of digestive cancer. The Japanese version of the Cancer Genome Atlas (JCGA) was used as the reference database [1]. Appendix A presents the signaling pathways associated with genomic alterations.

### 2.2. Statistical Analysis

The detection rate, sensitivity, specificity, and positive likelihood ratio were calculated using Microsoft Excel 2019 (Microsoft, Redmond, WA, USA). All statistical analyses were performed using IBM SPSS Statistics ver. 25 (International Business Machines Co., Armonk, NY, USA). The rate of characteristic genomic alterations was evaluated using the χ-square test or Fisher’s exact test, and *p*-values < 0.05 were considered statistically significant.

## 3. Results

Among the 1587 patients that underwent the FoundationOne CDx, 547 had digestive cancer. There were 333 male and 214 female patients and the median age at diagnosis was 62 years (range, 15–85) (Appendix A). Appendix A presents the alteration plots for each case. The primary sites of cancer were as follows: esophageal cancer (n = 27), gastric cancer (n = 43), duodenal cancer (n = 4), small intestine cancer (n = 6), colorectal cancer (n = 217), pancreatic cancer (n = 127), and biliary tract cancer (n = 123) (Appendix A).

The overall detection rates for potentially actionable genomic alterations, actionable genomic alterations, and genomic alterations equivalent to CDx for digestive cancers were 99.5% (544/547), 62.5% (342/547), and 11.5% (63/547), respectively (Figure 2 and Appendix A).

### 3.1. Detection Rate of Genomic Alterations in Digestive Cancers

Several genomic alterations were observed in the genes involved in the TP53 signaling pathway, mainly in *TP53*, in patients with digestive cancer. However, the genomic alterations and signaling pathways with the highest number of genomic alterations differed for each organ (Figure 3).

#### 3.1.1. Esophagus/Stomach Cancer

Histologically, 81.5% (22/27) and 14.8% (4/27) of esophageal cancers were classified as squamous cell carcinoma (SCC) and neuroendocrine carcinoma (NEC), respectively. The detection rates for potentially actionable genomic alterations, actionable genomic alterations, and genomic alterations equivalent to CDx were 100.0% (27/27), 85.2% (23/27), and 18.5% (5/27), respectively (Appendix A). MSI-H (0/27) was not observed in any of the cases; in contrast, TMB-H was observed in 14.8% (4/27) of cases (Appendix A). Several alterations were observed in the genes involved in the TP53 pathway, primarily in *TP53* (85.2%), as well as the cell cycle pathway, mainly in *CDKN2A* (59.3%) and *CDKN2B* (40.7%) (Appendix A).

Adenocarcinoma, accounting for 93.0% (40/43) of cases, was the most common histological type of gastric cancer. The detection rates for potentially actionable genomic alterations, actionable genomic alterations, and genomic alterations equivalent to CDx were 100.0% (43/43), 76.7% (33/43), and 30.2% (13/43), respectively (Appendix A). MSI-H was observed in 9.3% (4/43) of cases, whereas TMB-H was observed in 2.3% (1/43) of cases (Appendix A). Several alterations were observed in the genes in the TP53 pathway, mainly in *TP53* (69.8%), as well as the RTK pathway, mainly in *ERBB2* (27.9%) (Appendix A).

#### 3.1.2. Bowel Cancer

Duodenal, small intestinal, and colorectal cancers were grouped as bowel cancers. Adenocarcinomas accounted for all cases of duodenal cancers (4/4). The detection rates for potentially actionable genomic alterations, actionable genomic alterations, and genomic alterations equivalent to CDx were 100.0% (4/4), 100.0% (4/4), and 0.0% (0/4), respectively (Appendix A). MSI-H or TMB-H (0/4) was not detected in any of the cases (Appendix A). Several alterations were observed in genes involved in the TP53 pathway, mainly in *TP53* (75%) (Appendix A).

Adenocarcinoma, accounting for 66.7% (4/6) of cases, was the most common histological type of small intestine cancer. The detection rates for potentially actionable genomic alterations, actionable genomic alterations, and genomic alterations equivalent to CDx were 100% (6/6), 66.7% (4/6), and 0.0% (0/6), respectively (Appendix A). MSI-H or TMB-H (0/6) was not observed in any of the cases (Appendix A). Several alterations were observed in the genes involved in the Ras/Raf/MEK/ERK pathway, mainly in *KRAS* (33.3%) and *NRAS* (33.3%) (Appendix A).

Adenocarcinoma, accounting for 96.3% (209/217) of cases, was the most common histological type of colorectal cancer. The detection rates for potentially actionable genomic alterations, actionable genomic alterations, and genomic alterations equivalent to CDx were 99.5% (216/217), 51.6% (112/217), and 12.0% (26/217), respectively (Appendix A). MSI-H was observed in 0.9% (2/217) of cases, whereas TMB-H was observed in 5.1% (11/217) of cases (Appendix A). Several alterations were observed in the genes involved in the TP53 pathway, mainly in *TP53* (81.1%), as well as the Wnt/β-catenin pathway, mainly in *APC* (78.8%) (Appendix A).

#### 3.1.3. Pancreatic Cancer

Adenocarcinoma, accounting for 90.6% (115/127) of cases, was the most common histological type of pancreatic cancer. The detection rates for potentially actionable genomic alterations, actionable genomic alterations, and genomic alterations equivalent to CDx were 99.2% (126/127), 59.1% (75/127), and 2.4% (3/127), respectively (Appendix A). MSI-H was not observed in any of the cases (0/127); in contrast, TMB-H was observed in 0.8% (1/127) of cases (Appendix A). Several alterations were observed in the genes involved in the TP53 pathway, mainly in *TP53* (69.3%), as well as the Ras/Raf/MEK/ERK pathway, mainly in *KRAS* (87.4%) (Appendix A).

#### 3.1.4. Biliary Tract Cancer

Adenocarcinoma, accounting for 96.7% (119/123) of cases, was the most common histological type of biliary tract cancer. The detection rates for potentially actionable genomic alterations, actionable genomic alterations, and genomic alterations equivalent to CDx were 99.2% (122/123), 74.0% (91/123), and 13.0% (16/123), respectively (Appendix A). MSI-H was observed in 1.6% (2/123) of cases, whereas TMB-H was observed in 7.3% (9/123) of cases (Appendix A). Several alterations were observed in the genes involved in the TP53 pathway, mainly in *TP53* (48.8%), as well as the cell cycle pathway, mainly in *CDKN2A* (31.7%) and *CDKN2B* (21.1%) (Appendix A).

### 3.2. Genomic Alteration of Digestive Adenocarcinomas

Adenocarcinoma was the most common histologic type of digestive cancer, except in esophageal cancer. The detection rates for signaling pathways with genomic alterations (Appendix A) and overall genomic alterations (Appendix A) were analyzed for each organ. However, no significant differences were observed between the detection rates for adenocarcinoma and the other histological types owing to the high proportion of adenocarcinomas in each organ (Figure 3 and Figure 4).

Alterations in the genes involved in the RTK system, such as *ERBB2* amplification, were often observed in gastric adenocarcinoma. Alterations in the genes involved in the Wnt/β-catenin pathway, mainly *APC*, were observed in colorectal adenocarcinoma. Alterations in *KRAS*, which plays a role in the Ras/Raf/MEK/ERK pathway, were observed in pancreatic adenocarcinoma. Alterations in the genes involved in the cell cycle pathway, mainly in *CDKN2A* and *CDKN2B*, were frequently observed in biliary tract adenocarcinoma. Adenocarcinomas of the duodenum and small intestine showed no clear trend owing to the small number of cases analyzed.

The genes with numerous alterations in digestive adenocarcinomas, excluding duodenal and small intestinal adenocarcinomas, were cross-organized and evaluated across different types of digestive organs. The frequency of *APC* alterations in colorectal adenocarcinoma (80.9%) was significantly higher than that in stomach (2.5%), pancreatic (1.7%), and biliary tract (1.7%) adenocarcinomas (*p* < 0.01). The frequency of *KRAS* alterations in pancreatic (92.2%) and colorectal (45.5%) adenocarcinomas was significantly higher than that in the stomach (20.0%) and biliary tract (21.2%) adenocarcinomas (*p* < 0.01). The frequency of *CDKN2A* alterations in pancreatic (47.8%) and biliary tract (31.1%) adenocarcinomas was significantly higher than that in stomach (17.5%) and colorectal (1.4%) adenocarcinomas (*p* < 0.01) (Appendix A).

### 3.3. Diagnostic Flowchart of Digestive Adenocarcinomas

A diagnostic flowchart of genomic alterations in digestive adenocarcinomas was created based on these results (Figure 5). *APC*, *KRAS*, and *CDKN2A* were focused on in the present study, and cases with adenocarcinomas were classified based on the presence or absence of each alteration. Among the adenocarcinomas with *APC* alterations, 97.1% were colorectal adenocarcinomas (Appendix A). Colorectal (40.6%) and pancreatic (45.3%) adenocarcinomas were the most common digestive adenocarcinomas with *KRAS* alterations (Appendix A). Pancreatic (53.9%) and biliary tract (36.3%) adenocarcinomas were the most common digestive adenocarcinomas with *CDKN2A* alterations (Appendix A). Pancreatic adenocarcinoma (67.5%) was the most prevalent digestive adenocarcinoma without *APC* alteration but with *KRAS* alteration (Appendix A). Biliary tract adenocarcinoma (75.0%) was the most prevalent digestive adenocarcinoma without *APC* and *KRAS* alteration but with *CDKN2A* alteration (Appendix A). The rates of digestive adenocarcinoma without *APC*, *KRAS*, and *CDKN2A* alterations in patients with biliary tract, gastric, colorectal, and pancreatic adenocarcinomas were 54.8%, 21.0%, 18.5%, and 5.6%, respectively (Appendix A). A diagnostic flowchart was created and the sensitivity, specificity, and positive likelihood ratio were calculated to validate the usefulness of the diagnostic flow. The sensitivity, specificity, and positive likelihood ratio were calculated for gastric (65.0%, 77.8%, and 2.9, respectively), colorectal (80.9%, 98.2%, and 44.3, respectively), pancreatic (90.4%, 86.4%, and 6.6, respectively), and biliary tract cancers (20.2%, 97.8%, and 9.2, respectively) using this flowchart (Appendix A). Thus, *APC* alterations were highly associated with colorectal cancer; *KRAS* alterations and normal *APC* were highly associated with pancreatic cancer; and *CDKN2A* alterations and normal *APC* and *KRAS* were highly associated with biliary tract cancer.

## 4. Discussion

We established a characteristic genomic profile for each organ by separating actionable genomic alterations from those with minimal impact on carcinogenesis and VUSs using the scoring system developed in the present study. Notably, this profile was comparable with those reported in previous studies [21,22,23,24]. Evaluating CGP results using this scoring system may elucidate the genomic profile of each case. Subsequent comparison with the genomic profile of each organ established in the present study could help identify the primary site of unknown primary cancers.

In esophageal cancers, *CDKN2A*, *CDKN2B*, and *CCND1* alterations in the cell cycle pathway were frequently observed (Appendix A). A previous study reported *CCND1* amplification in the cell cycle pathway, and also *TP63/SOX2* amplification and *KDM6A* deletion in transcriptional regulation [25]. In gastric cancers, *ERBB2* alterations in the RTK pathway were observed. In a previous study, gastric cancers were classified into four categories—EBV (Epstein–Barr virus), MSI (microsatellite instability), CIN (chromosomal instability), and GS (genomically stable). Of these, CIN was characterized by *TP53* mutation, *ERBB2* amplification, *VEGFA* amplification, and RTK-RAS activation [26]. In colorectal cancers, *APC* alterations in the Wnt/β-catenin pathway were mainly observed. In a previous study, colorectal cancers were classified into four categories—CMS1 (MSI immune), CMS2 (canonical), CMS3 (metabolic), and CMS4 (mesenchymal). CMS2 was characterized by WNT and MYC activation and high SCNAs (somatic copy number alteration) [27]. In the present study, CMS2 was probably the most common. In pancreatic cancers, *KRAS* alterations in the Ras/Raf/MEK/ERK pathway were mainly observed. Molecular genomic analyses revealed that pancreatic ductal adenocarcinoma (PDAC) was composed of *KRAS, TP53, SMAD4,* and *CDKN2A*. In particular, *KRAS* alterations were found in over 90% of PDACs [28]. In biliary tract cancers, *CDKN2A* and *CDKN2B* alterations in the cell cycle pathway were observed; however, these are not effective therapeutic target genes. In a previous study, biliary tract cancers were classified according to druggable genes—FGFR (FGFR pathway alterations or *FGFR2* fusion/rearrangement), HER2 (*ERBB2* amplification and mutations), IDH1 (*IDH1* alterations), BRAF (*BRAF* alterations), MSI (MSI-H or MSI-deficient mismatch repair), NTRK (*NTRK* fusion/rearrangement), and others [29]. Although trastuzumab is expected to be an effective molecular target drug for *ERBB2* alterations, which were observed in gastric cancers, there are no effective drugs currently available for *KRAS* and *APC* alterations, which are frequently observed in other organs, or for *TP53* alterations, which are frequently observed in digestive cancers. In the future, more drugs would need to be developed.

Nevertheless, more than before, the treatment landscape for cancer is rapidly evolving owing to the increase in the number of approved drugs targeting specific genomic modifications [30,31]. In the present study, actionable genomic alterations were observed in 62.5% of all digestive cancers. Moreover, genomic alterations equivalent to CDx were observed in 11.5% of patients, demonstrating the clinical utility of genomic panel testing. A previous study, which analyzed advanced solid tumors using FoundationOne CDx or FoundationOne Heme, revealed one or more alterations in 94.6% of cases, as well as actionable alterations with candidates for therapeutic agents in 87.7% of cases [12]. Several clinical trials on CGP in patients with advanced or metastatic solid tumors have reported that the prevalence of actionable genomic alterations per patient ranges from 40% to 94% [32,33]. Furthermore, compared with previous reports, the percentage of patients with treatment recommendations ranged from 11% to 39% [13,34,35,36,37]. However, the rate of genotype-matched therapy was 9.4% in Japan (between June 2019 and June 2022) [38]. This is due to the Japanese healthcare insurance system, which permits the use of some molecular-targeted drugs and immune checkpoint inhibitors only if CDx-positive. In our study, the alterations observed in several patients with esophageal cancer (actionable, 85.2%; CDx-equivalent, 18.5%), gastric cancer (actionable, 76.7%; CDx-equivalent, 30.2%), and biliary tract cancer (actionable, 74.0%; CDx-equivalent, 13.0%) were associated with therapeutic agents. This finding suggests that this subset of patients would benefit from the active use of CGP. CGP is available only after the completion or expected completion of standard therapy in most cases in Japan [34]. However, the adoption of CGP during first-line chemotherapy can aid in therapeutic decision making. Furthermore, obtaining a genomic profile at the time of pre-treatment, especially when making a pathological diagnosis, would allow molecular classification, which would provide information on the nature of the cancer and prognostic prediction.

A diagnostic flowchart of genomic alterations in digestive adenocarcinomas was created based on our results (Figure 5). *APC*, *KRAS*, and *CDKN2A* were focused on in the present study, and cases with adenocarcinomas were classified based on the presence or absence of each alteration. The sensitivity, specificity, and positive likelihood ratio were high for colorectal adenocarcinoma. Similarly, the specificity was high and the positive likelihood ratio was relatively high for biliary tract adenocarcinoma. Thus, the established diagnostic flowchart was considered useful for the diagnosis of colorectal and biliary tract adenocarcinomas.

Nevertheless, the present study has some limitations. First, the definition of actionable alterations remains controversial. Second, the progress of the disease was not followed up. Third, although insurance requirements stipulate that the test should be performed after or at the time of completion or expected completion of standard treatment, in reality, the test may be performed at any time point at the discretion of clinicians [39]. Fourth, various types of tissue, such as endoscopic ultrasound–fine needle aspiration (EUS-FNA) or operative tissue, were used in the present study. Therefore, it is possible that not all genomic alterations were detected in cases with a low tumor content. Fifth, the tissue specimens were subjected to treatments that may have altered the true genomic profiles of each digestive cancer. As the specimens were obtained at various time points before and after chemotherapy and belonged to both primary tumors and metastases, it is difficult to discern whether they were from a pure genomic profile of digestive cancer. Lastly, the flowchart developed in the present study has only been validated for digestive adenocarcinomas, but not adenocarcinomas of other organs. Thus, the usefulness of the flowchart must be validated using the CGP results of metastatic tissue with known primary sites. Furthermore, it is expected that the flowchart will become more practical by evaluating the results of liquid biopsy. In the future, a flowchart including all organ adenocarcinomas can be used to identify the primary site of an unknown primary cancer. Despite these limitations, the findings of the present study, along with the proposed diagnostic flowchart based on the genome analysis of real-world data, add unique value to clinical practice owing to their versatility, practicality, and suitability.

## 5. Conclusions

This large-scale study assessed the utility of CGP in diagnosing digestive cancers and proposed a diagnostic flowchart based on gene alterations. The detection rates of actionable genomic alterations were high across various digestive cancer types. We believe that a comprehensive diagnosis based on histopathological images and CGP will aid in the precise diagnosis and treatment of cancer. Further prospective clinical trials assessing the overall survival and quality of CGP-based targeted therapies must be conducted to aid in decision making in the era of personalized treatment.

## Figures and Tables

**Figure 1 cancers-16-01504-f001:**
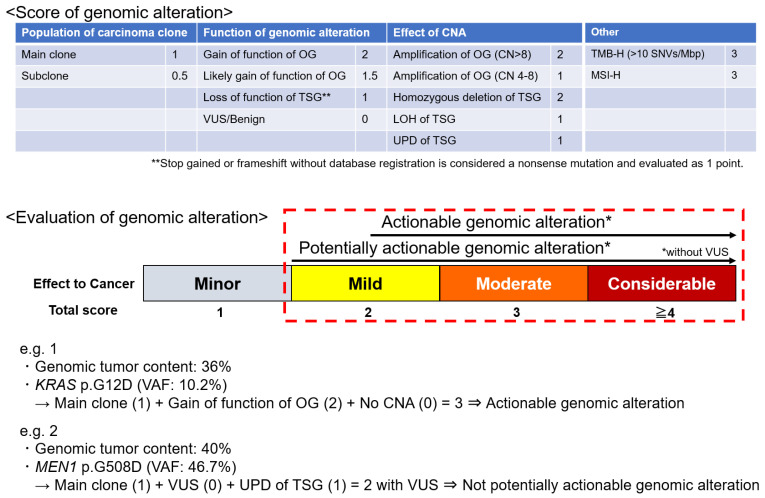
Scoring system of genomic alteration. A score is assigned to each item based on its impact on carcinogenesis. The sum of the scores of the population of carcinoma clones, the function of genomic alteration, and the effect of CNA is the score for that alteration. TMB and MSI are scored as high, in addition to individual genomic alterations. Potentially actionable genomic alteration with biological significance is defined by a score of ≥2 points without VUSs. An actionable genomic alteration candidate for drugs that may be useful is defined by a score of ≥2.5 points without VUSs. CN, copy number; CAN, copy number alteration; LOH, loss of heterozygosity; MSI-H, high microsatellite instability; OG, oncogene; TMB-H, high tumor mutation burden; TSG, tumor suppressor gene; UPD, uniparental disomy; VUS, variant of uncertain significance.

**Figure 2 cancers-16-01504-f002:**
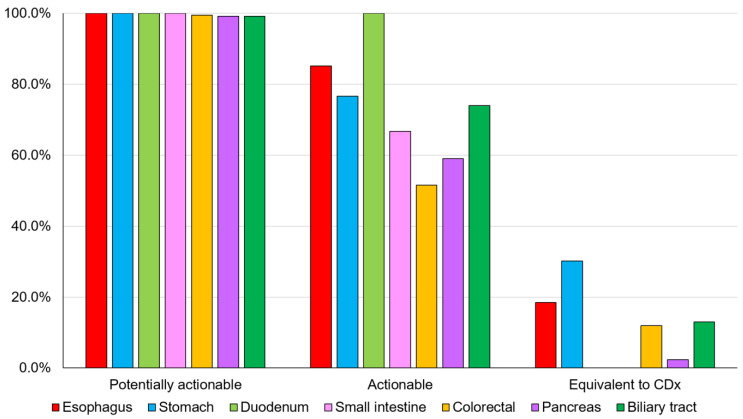
Detection rate of genomic alterations. The detection rates for potentially actionable genomic alterations, actionable genomic alterations, and CDx-equivalent genomic alterations for each digestive cancer (Appendix A). CDx, companion diagnostic.

**Figure 3 cancers-16-01504-f003:**
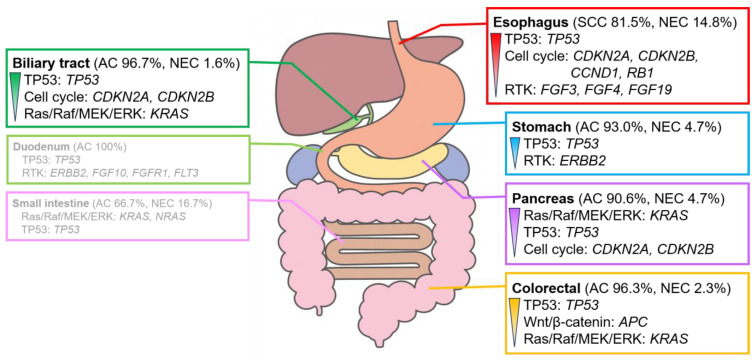
Characteristic signaling pathways with genomic alterations in digestive cancers. Signaling pathways with genomic alterations of >30% in each digestive cancer. The pathways are listed in the order of their prevalence. Genomic alterations are observed in >20%. The duodenum and small intestine show no clear trend owing to the small number of cases analyzed. For further details, please refer to Appendix A. AC, adenocarcinoma; NEC, neuroendocrine carcinoma; SCC, squamous cell carcinoma.

**Figure 4 cancers-16-01504-f004:**
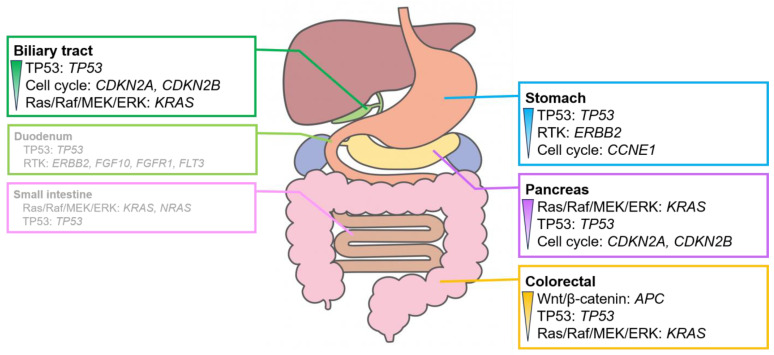
Characteristic signaling pathways with genomic alterations in digestive adenocarcinomas. Signaling pathways with genomic alterations exceeded 30% in digestive adenocarcinomas. The pathways are listed in order of their prevalence. Genomic alterations are observed in over 20% of cases. The duodenum and the small intestine exhibit no clear trend owing to the small number of cases analyzed. For further details, please refer to Appendix A.

**Figure 5 cancers-16-01504-f005:**
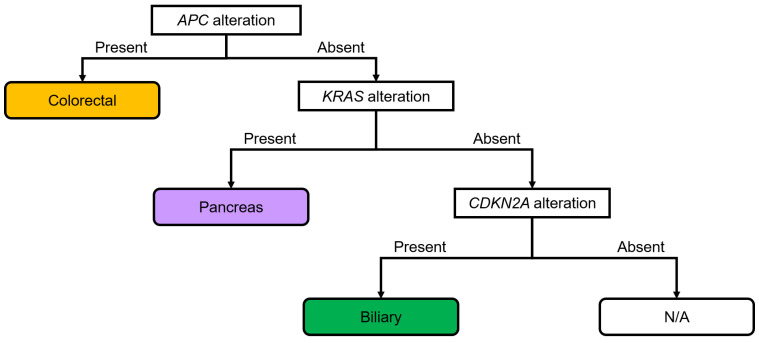
Diagnostic flowchart based on the genomic alterations in digestive adenocarcinomas. *APC* alterations show strong associations with colorectal cancer. *KRAS* alterations and lack of alterations in *APC* show strong associations with pancreatic cancer. A lack of alterations in *APC* and *KRAS*, but the presence of alterations in *CDKN2A*, shows strong associations with biliary tract cancer. The sensitivity, specificity, and positive likelihood ratios were calculated for each cancer type (Appendix A). N/A, not applicable.

## Data Availability

All data generated or analyzed during this study are included in this published article and the Appendix A.

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
