# Peer review of "Clinical and Diagnostic Utility of Genomic Profiling for Digestive Cancers: Real-World Evidence from Japan"

_cancers, 2024, doi:10.3390/cancers16081504_

Round 1

Reviewer 1 Report

Comments and Suggestions for Authors

The authors discussed the Clinical and diagnostic utility of genomic profiling for  digestive cancers. These are the important findings to add to the present available scientific literature but, there are several other components to be addressed before it could be considered for possible publication.

Revisions:

1.     In abstract section line “74-75”, does authors mean APC, KRA and CDKN2A are actionable for adenocarcinoma’s only.

2.     Introduction needs further improvement to clarify the aim of the study. “The adoption of CGP during first-line chemotherapy can aid in therapeutic decision-making; however, CGP is available only after the completion or expected completion of standard therapy in most cases in Japan”, clarifies the sentence with respect to the importance of the current study. What is the importance of the approval of insurance of CPG testing in the current study, it is not advisable to write it in the introduction.

3.      Please clarify the importance of para “Previous studies ------------- in cancers remain underexplored” for executing the current study in the same setup.

4.     In Methodology line 123-124 “patients with cancer were recruited in this study”? also reframe the sentence “Figure S1 presents the workflow used for cancer genome testing by the Keio PleSSision Group” line  134.

5.     In the result section, “Among the 1587 patients included in this study, 547 had digestive cancer. Figure S2 194 presents the alteration plots for each case”. -why other than digestive cancers were recruited, in methodology explanation on subject selection is missing.

6.     The discussion part is too concise and needs to be further strengthened concerning result findings, at least the main findings need explanation.

7.     Overall organization lacks clarity. In some places, the sentences are not in continuity, which makes it hard to interpret the manuscript content.

Comments on the Quality of English Language

NA

Reviewer 2 Report

Comments and Suggestions for Authors

The authors present valuable data on a highly applicable topic and the paper is mostly clearly written. Overall, the paper is informative, I only have a few suggestions to improve clarity of some of the sections.

The authors stated that diagnosing digestive cancer was one of their aims, but the test applied is intended for tissue analysis. Is there a version of the platform they used intended for liquid biopsy? The authors should discuss more clearly the strategy that they propose to be used for future diagnostic purposes.

Also related to the methodology, it would be beneficial to include a couple of sentences on the basic principle of FoundationOne CDx, as a specific platform used in this research. The current formulation given in line 102 is unclear. Additionally, advantages and limitations of the technology should be discussed, and also compared between the platform intended for tissue vs. the liquid biopsy version.

I would expect the authors to include at least basic demographic data on patients included in the study.

It is unclear what the authors meant by the sentence in lines 178-179: “Genomic alterations equivalent to CDx” were defined as genomic alterations caused by drugs under insurance coverage in Japan. Causal inference seems to be unadequate here.

Line 371-372 states that „Subsequent comparison with the genomic profile of each organ established in the present study could help identify the primary site of unknown primary cancers“. Again, the use of the studied platform for diagnostic purposes would require the version of the technology intended for liquid biopsy. If the authors suggest the use of the platform not only predictive, but also for diagnostics or patient stratification purposes, than this should be explained more clearly.

Comments on the Quality of English Language

There are a couple of unclear formulations (listed in the comments)
